# Dual-Polarized Fiber Laser Sensor for Photoacoustic Microscopy

**DOI:** 10.3390/s19214632

**Published:** 2019-10-24

**Authors:** Xiangwei Lin, Yizhi Liang, Long Jin, Lidai Wang

**Affiliations:** 1Department of Biomedical Engineering, City University of Hong Kong, 83 Tat Chee Ave, Kowloon 999077, Hong Kong, China; lin_xiangwei@yeah.net; 2City University of Hong Kong Shenzhen Research Institute, Yuexing Yi Dao, Nanshan District, Shenzhen 518057, China; 3Guangdong Provincial Key Laboratory of Optical Fiber Sensing and Communications, Institute of Photonics Technology, Jinan University, Guangzhou 510632, China; liangyizhi88528@gmail.com (Y.L.); iptjinlong@gmail.com (L.J.)

**Keywords:** fiber laser sensor, dual polarization, high sensitivity, high stability, optical resolution photoacoustic microscopy

## Abstract

Optical resolution photoacoustic microscopy (OR-PAM) provides high-resolution, label-free and non-invasive functional imaging for broad biomedical applications. Dual-polarized fiber laser sensors have high sensitivity, low noise, a miniature size, and excellent stability; thus, they have been used in acoustic detection in OR-PAM. Here, we review recent progress in fiber-laser-based ultrasound sensors for photoacoustic microscopy, especially the dual-polarized fiber laser sensor with high sensitivity. The principle, characterization and sensitivity optimization of this type of sensor are presented. In vivo experiments demonstrate its excellent performance in the detection of photoacoustic (PA) signals in OR-PAM. This review summarizes representative applications of fiber laser sensors in OR-PAM and discusses their further improvements.

## 1. Introduction

Optical resolution photoacoustic microscopy (OR-PAM) is a prosperous and growing biomedical imaging modality that provides high-resolution, non-invasive, and label-free functional imaging of healthy and diseased tissues [1,2,3]. Based on the photoacoustic (PA) effect, nanosecond laser pulses induce ultrasonic waves. Thus, an acoustic sensor should be able to detect the PA wave with high sensitivity, broad bandwidth, wide acceptance angle, and high stability [4,5,6]. Besides, the acoustic detection beam employed in OR-PAM needs to be aligned with the focused optical excitation beam to achieve high sensitivity and high resolution for in vivo imaging [7,8,9,10,11,12]. Most OR-PAM techniques use piezoelectric detectors to receive PA signals [13,14,15,16]. A piezoelectric detector may suffer from low sensitivity with reduced sensor size. Additionally, it is complicated to deliver a laser beam in a confined space such as an endoscope. Therefore, the development of a new photoacoustic detector is of urgent demand in photoacoustic microscopy.

Optical photoacoustic detection has been developed in recent years [13,17,18,19,20,21,22,23,24,25,26]. Compared with piezoelectric detectors, optical photoacoustic detectors usually possess high sensitivity. For example, a micro-ring resonator-based photoacoustic sensor has a 105 Pa noise-equivalent pressure (NEP) over a bandwidth of 280 MHz [18]. A planar Fabry–Perot polymer film has a 210 Pa NEP with 20 MHz bandwidth [19,20]. A two-wave mixing interferometer detects PA signals with a maximum bandwidth of 200 MHz [21,22]. Photoacoustic remote sensing microscopy with non-interferometric architecture achieves a measured signal to noise ratio (SNR) of 60 dB with 2.7 ± 0.5 μm spatial resolution [23,24]. A glass substrate-based gold nanostructure etalon reaches 40 MHz center frequency with a bandwidth of 57 MHz [25]. The wide bandwidth and high sensitivity of an optical ultrasound sensor dramatically enhances the performance of OR-PAM. However, the remaining challenge is that high sensitivity may cause poor resistance to external mechanical or thermal disturbances. 

Besides the aforementioned sensors, optical fiber lasers are an emerging technique for PA detection in OR-PAM [27,28,29,30,31]. The sensitivity of the fiber laser sensor does not decrease as its size is reduced, making it perfectly match the miniaturization case. To the best of our knowledge, three types of optical fiber-based ultrasound sensors have been developed for photoacoustic imaging. The first one is a Fabry–Perot resonator on the fiber tip to detect the PA signal. The sensor can provide a NEP of 68.7 Pa over 80 MHz bandwidth. In vivo imaging of a mouse ear with a 10 × 10 mm^2^ field of view was demonstrated in Ref. [29]. The second type is a pi-phase-shifted fiber Bragg grating-based sensor. Different from the resonator on the fiber tip, the Bragg grating sensor has an in-fiber cavity and thus can detect ultrasound in the radial direction of the fiber. A NEP of 440 Pa was achieved with 10 MHz bandwidth. Photoacoustic imaging was demonstrated via in vivo mouse ear imaging and ex vivo intravascular imaging [30,31]. The Fabry–Perot resonator sensor and the Bragg grating-based sensor have been systematically reviewed [13,19]. Here, we focus on reviewing a recently developed dual-polarization fiber-laser sensor for photoacoustic microscopy.

Considering the detection sensitivity, bandwidth, sensor size, and detection field of view [32,33,34,35,36,37,38,39,40,41,42], fiber laser sensors based on the dual-polarization principle have their unique advantages in OR-PAM. A fiber-laser-based ultrasound sensor can achieve a 40 Pa NEP over a 50 MHz bandwidth [43,44,45,46]. The smaller size makes it straightforward to combine the acoustic detection beam with the optical excitation beam. The cylindrical geometry and side-looking ability make it suitable for PA endoscopy or wearable devices. In addition, the differential detection between two polarizations ensures excellent stability while maintaining high detection sensitivity. Here, we review recent progress on the dual-polarization-based fiber laser sensor and its applications in OR-PAM. We first present the sensor principle, including sensor design, fabrication, signal demodulation, and noise analysis. We then discuss the characterization and optimization of sensitivity. The last part summarizes the in vivo application of the sensor in OR-PAM.

## 2. Principle of the Dual-Polarized Fiber Laser Sensor

### 2.1. Fabrication and Sensing Principle

A fiber laser with a short cavity and orthogonally polarized mode emits monochromatic light. Once an ultrasonic wave exerts pressure on the radial direction of the fiber laser cavity, the resonant frequencies of the two polarized modes change differently, and thus the beat frequency varies with ultrasonic pressure. A schematic of the dual-polarization fiber laser sensor is shown in Figure 1. The sensor was fabricated with an Er/Yd co-doped fiber (Er/Yb codoped fiber, EY305, CorActive, Canada). We photo-inscribed two wavelength-matched, highly reflective intracore Bragg gratings in the fiber core to form a resonant cavity. A 193 nm ArF excimer laser with a 1059 nm pitch phase mask was used to fabricate above gratings of length *L_g_* and grating separation *L_s_*. The two gratings and the gain medium between them formed the Fabry–Perot cavity of the fiber laser. To ensure the single longitude mode operation, the grating separation *L_s_* was typically less than 1 cm. Fiber absorption of the pump wavelength (980 nm) was ~1337 dB/m, which offered high gain to the fiber laser. Each grating had a coupling strength of ~25 dB, providing strong optical reflectivity. The length of one grating *L_g_* was 3.0 mm. After fabricating the gratings, the annealing process of the fiber laser was performed for 120 min at 120 °C to reduce the photon darkening effect induced by UV exposure. 

The lasing frequency is determined by the resonant cavity. The laser emits two linear polarization modes due to the weak birefringence of optical fiber. Each polarization mode can be expressed by [44,45]:(1)$$\frac{4\pi }{c}f_{x,y}\int_{-\infty }^{+\infty }n_{x,y}\left(z\right){\left|e\left(z\right)\right|}^{2}dz=2M\pi$$
where *c* is the light speed in vacuum, $f_{x,y}\;$is the lasing frequency for each polarization mode *x* and *y*, $n_{x,y}$ is the refractive index, and *M* denotes the resonant order. The term ${\left|e\left(z\right)\right|}^{2}$represents the longitudinal profile of intracavity intensity, which is normalized as $\int_{-\infty }^{+\infty }{\left|e\left(z\right)\right|}^{2}dz=1$. When the intensity of the two polarization modes are detected by the photodetector, the subtle difference between lasing frequency yields a beat signal at radio-frequency (RF) range. The beat frequency can be expressed as:(2)$$\Delta f=\frac{c}{n_{0}\lambda }\int_{-\infty }^{+\infty }B\left(z\right){\left|e\left(z\right)\right|}^{2}dz$$
where $B\left(z\right)=\left(\left|n_{x}-n_{y}\right|\right)$ is the local birefringence. When the fiber is free from perturbation, the birefringence is mainly caused by imperfection. For example, the fiber cross section may deviate from a perfect circular geometry. Thermal stress and resultant non-uniform strain in the fiber core may induce additional birefringence. At the grating regions, the local birefringence may be affected by the UV side illumination in photo inscribing. Thus, each fiber laser has a unique beat frequency, even fabricated with the same fiber. The principle axis of the fiber is unknown during fabrication of the fiber laser. As heat can change birefringence, we used a CO_2_ laser to irradiate the fabricated fiber laser, so that we could finely tune the beat frequency to the preassigned value. 

The fiber laser sensing system is illustrated in Figure 2. A continuous-wave laser diode (980 nm) pumped the fiber laser through a wavelength division multiplexer (WDM). The pump power was dozens of milliwatt to maximize the laser output. An optical isolator was connected to the fiber laser output to prevent back reflections that makes the fiber laser unstable. The two polarization modes were orthogonal. To detect the beat signal, a linear polarizer was used to project the two orthogonal polarization modes into the same axis. A polarization controller adjusted the laser polarization states to maximize the beat signal, where its frequency and intensity could be stabilized by the polarization-maintaining fiber. The laser output power was about 0.5–1 mW. Then, we amplified the power to 28 mW using an erbium-doped fiber amplifier so that the SNR at the photodetector could be increased. The beat signal from the photodetector (DSC40S, Discovery, Ewing, NJ, USA) was measured with a vector signal analyzer to determine the frequency.

As shown in Figure 3a, the two polarization modes can be visualized in the optical spectrum (BOSA 200 CL, Aragon Photonics Labs, Zaragoza, Spain), and these two modes output almost the same power. Considering the polarization-burning-hole effect, the mode competition in the fiber laser is negligible. Figure 3b illustrates the stably detected beat signal at the beat frequency of 2.74 GHz, where its intrinsic birefringence is 2.05 × 10^−5^. From Equation (2), the change of fiber birefringence leads to beat frequency variation. Hence, the fiber laser sensor was a birefringence sensor. When the optical fiber engages the acoustic pressure, it is compressed along the ultrasound incident direction. The birefringence of the fiber laser is changed by the acoustic pressure in the radial direction, which subsequently induces the frequency shift in the beat signal [47]. As this frequency shift is proportional to the acoustic pressure, the PA signals can be recovered via a subsequent frequency demodulation procedure.

### 2.2. Signal Demodulation and Noise Analysis

The fiber laser sensor used in this work presented the frequency shift of the beat signal in response to incident acoustic waves. To recover the acoustic pressure, a frequency modulation and demodulation system based on I/Q frequency demodulation was required. The RF beat signal (carrier frequency *f_c_* = ~2.74 GHz) from the photodetector was connected to a vector signal analyzer (Pxie-5646R, NI, Austin, TX, USA), where two low-noise quadrature signals were mixed with the modulated RF signal. The frequencies of the quadrature signals were close to the carrier frequency of the beat signal. After signal mixing and low-pass filtering, the I and Q quadrature signals were able to extract the phase φ of the beat signal. Then, the frequency was deduced as $f_{b}=d\varphi /dt$. The sampling rate for the I and Q quadrature signals was 100 MHz in the experiments, which allowed for acquiring PA signal with a 50 MHz bandwidth. 

Our fiber laser sensor shared similar noise sources as a microwave photonics system [44,45]. The total noise *n*_0_ mainly originates from the fiber laser *n**_sen_*, optical amplifier *n*_edfa_, photodetector *n*_pd_ and data acquisition *n*_acq_. The photodetector noise includes thermal noise *n*_th_ and shot noise *n*_sh_. Each noise term depends differently on the optical power *P*_opt_. The total noise can be expressed as *n*_0_ = *n*_sen_ + *n*_edfa_ + *n*_th_ + *n*_sh_ + *n*_acq_. Each noise term can be written as: (3)$$\begin{array}{c} n_{sen}=k_{1}P_{opt}^{2},n_{edfa}=k_{2}P_{opt}^{2},\\ n_{sh}=k_{3}P_{opt},n_{th}=k_{4}T,n_{acq}=k_{5} \end{array}$$
where T is the absolute temperature, and *k*_1_, *k*_2_, *k*_3_, *k*_4_, *k*_5_ are constant coefficients.

The SNR of the demodulated signal can be written as:(4)$$\Gamma =\frac{1}{\left(\Delta f_{noise}\right)}=\sqrt{\frac{3k_{0}P_{opt}^{2}}{2nB^{3}}}=\frac{1}{B^{\frac{3}{2}}}\sqrt{\frac{3k_{0}P_{opt}^{2}}{2(k_{1}+k_{2})P_{opt}^{2}+k_{3}P_{opt}+k_{4}T+k_{5}}}$$
where *B* is the measurement bandwidth, $P_{opt}$ is the RF signal power, and *k*_0_ is the photon-to-electron conversion efficiency [48,49]. Using measurement data in Figure 4, the coefficients in Equations (3,4) are calculated as *k*_1_ = 5.8 × 10^−14^, *k*_2_ = 7.8 × 10^−14^, *k*_3_ = 1.6 × 10^−17^, *k*_4_ = 1.6 × 10^−20^, and *k*_5_ = 5.54 × 10^−19^. Both thermal noise and shot noise at 1 mW are −177 dBc/Hz. From *k*_1_, the noise of the fiber laser sensor is −145dBc/Hz, and the noise of the EDFA is ~6 dB. The fitted SNR curve is plotted as a dashed-dot line in Figure 4. In Zone 1, where we have low optical power (*P*_opt_ < 1 mW) on the photodetector, the SNR is mainly limited by thermal noise and is almost proportional to *P*_opt_. In Zone 2 (1 mW < *P*_opt_ < 3 mW), shot noise is the dominant noise source. The SNR is approximately proportional to *P*_opt_^1/2^. When further increasing the optical power, both thermal and shot noises become less significant, and the SNR becomes stable (labeled as ‘Zone 3’). When the input power exceeds the saturation power of the photodetector, the SNR may decrease due to reduced photodetector efficiency (*k*_0_ in Equation (4)). 

## 3. Characterization and Optimization of Sensitivity

### 3.1. Frequency Response

The fiber sensor had a cylindrical shape. The pressure-induced deformation was able to be calculated with a vector acoustic scattering model. The scalar solution cannot describe the axially asymmetric modes, i.e., modes with nonzero azimuthal orders in the context, which are typically mixed with shear/longitudinal waves. Instead, a general model was applied to describe the fiber frequency response. We first calculated the plane-wave case, which can be simplified as a two-dimensional problem in the fiber cross-section plane. The solutions of the scalar φ and vector potentials $\vec{H}$ are:(5)$$\varphi =\underset{l}{\sum }A_{n}J_{l}\left(k_{L}r\right)cos(l\theta )$$
(6)$$H_{z}=\underset{l}{\sum }B_{n}J_{l}\left(k_{S}r\right)sin(l\theta )$$
where *A_n_*, *B_n_* denote amplitudes of the potentials, *J_l_* is the lth order Bessel function, *k_S,L_*
*=* ω/*c_S,L_* is the wave number of the longitudinal or shear waves in the fiber. Here, only the frequency-domain response was considered, and the time-dependent factor exp(*iωt*) was ignored for simplification.

The displacement $\vec{u}\left(r,\theta \right)$ can be expressed as:(7)$$\vec{u}=-\nabla \varphi +\nabla \times \vec{H}$$

For free vibration, we have:(8)$$\left(\begin{array}{cc} a_{11} & a_{12}\\ a_{21} & a_{22} \end{array}\right)\left(\begin{array}{c} A_{n}\\ B_{n} \end{array}\right)=0$$
where the matrix elements are $a_{11}=Z_{L}^{2}\left(qJ_{n}\left(Z_{L}\right)-2J_{n}^{\prime \prime }\left(Z_{L}\right)\right)/Z_{S}^{2}$, $a_{12}=2n\left(J_{n}^{\prime }\left(Z_{s}\right)-J_{n}\left(Z_{S}\right)\right)/Z_{S}^{2}$, $a_{21}=2n\left(J_{n}^{\prime }\left(Z_{L}\right)-J_{n}\left(Z_{L}\right)\right)$, and $a_{22}=-J_{n}^{\prime \prime }\left(Z_{S}\right)+J_{n}^{\prime }\left(Z_{S}\right)-n^{2}J_{n}\left(Z_{S}\right)$. Z_L_ = k_L_a, Z_S_ = k_S_a, Z = ka, q = λ/μ, λ and μ are the Lamé elastic constants for compressibility and shear modulus, respectively.

The equations have nonzero solutions when its determinant $\left|\begin{array}{cc} a_{11} & a_{12}\\ a_{21} & a_{22} \end{array}\right|$ = 0, yielding a discrete spectrum of acoustic resonance. Here, the *l* = 2 modes were investigated because only these modes induced differential stresses between the *x* and *y* directions, causing birefringence variation. Each eigen-mode was denoted as (*l*, *n*), where *l* and *n* are the azimuthal and radial mode index, respectively. The resonant frequencies of first and second radial order modes were *f*_(2, 1)_ = 22.3 MHz and *f*_(2, 2)_ = 39.6 MHz. Their displacement profiles are plotted in Figure 5. For *l* = 0 mode, i.e., the axial symmetrical one, *B*_n_ = 0, only the compressional waves exist. Equation (8) degenerates as *a*_11_*A*_n_ = 0, which was used for mode calculations. When the fiber loses its axial symmetry via post-processing like side polishing, then mechanical modes with other azimuthal orders will be detected. Particularly, *l* = 2 modes with higher radial number are also simultaneously excited, but their resonant frequencies are beyond the detection bandwidth of frequency demodulation. 

Considering these acoustic eigen modes of silica fiber damped by the surrounding medium, the fiber vibration can exert pressure waves. The waves can be depicted as outwards propagating cylindrical waves $C_{n}H_{n}^{\left(1\right)}\left(kr\right)$, where $H_{n}^{\left(1\right)}$represents outwards propagating pressure and *C_n_* denotes its amplitude. The interaction between the solid fiber and the surrounding medium can be expressed as: (9)$$a_{11}A_{n}+a_{12}B_{n}+a_{13}C_{n}=0$$
where *a*_13_ is $H\left(Z\right)/\rho_{s}\omega^{2}$, and *b*_1_ equals zero. Though the shear waves are not supported in fluidic medium, the expression presenting zero shear stress at the boundary still holds, which can be rewritten as: (10)$$a_{21}A_{n}+a_{22}B_{n}=0$$

Also, the continuity of radial displacement demands:(11)$$a_{31}A_{n}+a_{32}B_{n}+a_{33}C_{n}=b_{3}$$
where $a_{31}=-J_{n}^{\prime }(Z_{L})$, $a_{32}=nJ_{n}(Z_{S})$ and $a_{33}=-H_{n}^{\left(1\right)\prime }(Z)/\rho_{s}\omega^{2}$, and b_3_ denotes the radial displacement created by the acoustic dipole source, which can be expanded as $b_{3}=\sum_{l}\left(b_{L,l}J_{l}\left(Z_{L}\right)\cos(l\theta )+b_{S,l}J_{l}\left(Z_{S}\right)\cos(l\theta )\right)$, where the subscript L and S denote the contributions from compressional and shear waves, respectively. Combining Equations (9)–(11), the coefficients *A_n_*, *B_n_*, and *C_n_* can be solved. 

The response in beat frequency shift Δ*f*_b_ is proportional to the birefringence change, i.e., $\Delta f_{b}=\frac{c}{n_{eff}\lambda }\Delta B$. The birefringence change Δ*B* is determined by: (12)$$\Delta B=-p_{44}n_{0}^{3}\left(k_{L}^{2}A_{n}-k_{S}^{2}B_{n}\right)/2=-p_{44}n_{0}^{3}A_{n}\left(k_{L}^{2}+k_{S}^{2}\frac{a_{31}}{a_{33}}\right)/2$$
where *n_eff_* means the effective index of the optical mode, *p_44_* means the photoelastic coefficient. The model was experimentally verified, with the calculated frequency response found to be consistent with the measured results. As shown in Figure 5b, the original two peaks at 22.3 MHz and 39.6 MHz broadens because of acoustic interaction with the surrounding medium. The (2, 1) and (2, 2) modes present significantly different 3-dB bandwidth. The measured sensitivity at 39.6 MHz is lower than theoretical calculation, which may be caused by water absorption. The acoustic pressure-induced fiber cross-sectional deformations at different response frequency are illustrated in Figure 5c,d. The indexes of the in-plane vibration modes (azimuthal and radial) are denoted as (*l*, *n*), and thus the above two frequency response peaks correspond to (2, 1) and (2, 2), respectively. The (2, 1) mode in Figure 5c indicates the compression of the fiber along the ultrasound incident direction, and the (2, 2) mode in Figure 5d corresponds to the case of stretching the outer region while compressing the inner region of the fiber. From the above theoretical analysis, the fiber frequency response is dependent on the cross-sectional size of the fiber whereby can be adjusted by adjusting the fiber diameter. Here, we used HF-etching to reduce the fiber diameter to ~60 μm. The center frequency was tuned to ~42 MHz, and the bandwidth was extended to ~20 MHz.

### 3.2. Spatial Sensitivity

The fiber sensitivity to planer acoustic waves was calibrated using an unfocused ultrasound transducer (V358-SU, Panametrics, USA). Pulsed ultrasound waves propagated normally to the fiber laser sensor. The aperture of ultrasonic wave was ~6 mm, comparable with the fiber sensing region of ~5 mm in length. The acoustic wave induced perturbation uniformly over the entire sensor. The measured temporal and frequency responses are shown in Figure 6, where 198 MHz beat frequency shift occurs at 88 kPa acoustic pressure. The acoustic sensitivity for the 60 μm fiber is 2.25 kHz/Pa, with 40 Pa NEP over a 50 MHz bandwidth. The acoustic sensitivity for the 125 μm fiber is 1.7 kHz/Pa, and the NEP is ~45 Pa with frequency peaking at ~22 MHz and ~39 MHz.

For photoacoustic microscopic imaging, the acoustic source is typically a point source. Thus, it was important to explore how the fiber sensor responded to a point source, i.e., the acoustic response at different positions (*r*, *θ*, *z*). Along the fiber direction, the fiber acts as an ideal line detector with cavity size *L*_c_, and the lasing frequency depends on the resonant condition in Equation (1) at each polarization $f_{x,y}$. The intracavity optical intensity density *e*(*z*), whose laser mode profile decides on the cavity length and the grating parameters, can weigh the sensitivity of the fiber laser sensor. The beat signal variation $\delta f_{b}$ caused by the local birefringence change $\delta B\left(z\right)$ can be written as:(13)$$\delta f_{b}=\frac{c}{n_{o}\lambda }\int_{-L/2}^{L/2}\delta B\left(\left|p\right|,\omega ,z\right){\left|e\left(z\right)\right|}^{2}\frac{exp(ik_{a}r)}{r}dz$$

For a spherical wave, the acoustic phase changes along the fiber due to different arrival times. The acoustic wave will be canceled out if the phase difference is beyond π. Thus, the sensitivity mainly originates from the perturbation accumulate over a confined region, where the phase is almost unchanged. As a result, Equation (13) can be approximated as:(14)$$\delta f_{b}=\frac{c}{n_{o}\lambda }L_{eq}\left(\omega \right)\delta B\left(\left|p\right|,\omega ,0\right){\left|e\left(z\right)\right|}^{2}$$
where $\delta B\left(\left|p\right|,\omega ,0\right)$ is the normal incident plane wave-induced birefringence change. We can see three determinative factors could contribute to the acoustic response: $\delta B\left(\left|p\right|,\omega ,0\right)$ depends on the fiber cross-sectional geometry and mechanical characters, ${\left|e\left(z\right)\right|}^{2}$ is the laser mode distribution, $L_{eq}\left(\omega \right)$ is the equivalent interaction length when considering phase cancellation. Figure 7 demonstrates that a point source *S* generates a spherical acoustic wave, which propagates along distance *d* to reach the line detector. The acoustic pressure is $p\left(\omega ,r\right)=\frac{\exp(ik_{a}r)}{r}$, where *ka* denotes the acoustic wave number, *r* = (*d*_2_ + *z*_2_)_1/2_ is the propagation length, and *z* is the longitudinal position. Based on the relationship, the equivalent interaction length was calculated as *L_eq_* = 2.506*d*/*ka*, which was much shorter than the effective sensing length, i.e,. 2–10 mm, of the fiber laser sensor. 

Figure 8 shows the measured frequency response of the fiber laser sensor. The acoustic source is located in the plane at a distance of *d* = 1 mm from the sensor, as shown in Figure 8a. Based on Equation (14), the frequency response decides on the product $L_{eq}\left(\omega \right)\delta B\left(\left|p\right|,\omega ,0\right)$. The effective length *L_eq_* remains unchanged for the same distance but different lateral positions. In the *x* direction, the fiber laser sensor works as a point detector, which also shows a nearly flatten frequency respond with the acoustic source at different positions. As shown in Figure 8c, the measured frequency responses along the *z* axis are unchanged. In addition, the variation of the amplitude at the peak frequency of 39 MHz comes from the increased loss at further distance.

The cavity behavior of the laser mode was determined by the laser mode distribution, ${\left|e\left(z\right)\right|}^{2}$, which was affected by the grating separation, fiber gain, and grating coupling strength. Between the two gratings, the forward and backward lights experience amplifications and reach maximum intensity before arriving at the gratings. As a result, the Fabry–Perot fiber laser typically presents a profile with two peaks at the inner grating edges. Fibers with higher gain can create sharper peaks and those with lower gain lead to flat-top profiles; thus, higher coupling strengths enable higher slopes. Based on the coupled-mode theory [50,51], the intensities rapidly decrease over the gratings with a simple relation T = 1 − tanh^2^(κz), thus the normalized intensity profile can be expressed by [52]:(15)$${\left|e\left(z\right)\right|}^{2}=\kappa \cdot e^{-2\kappa \left|z\right|}$$
where *κ* is the coupling coefficient of the gratings, and *z* represents the penetration depth. 

When the cavity laser is longer than 2 mm, it can be approximated as a Fabry–Perot laser. The corresponding rectangular mode profiles can be written as:(16)$${\left|e\left(z\right)\right|}^{2}=\left\{\begin{array}{cc} \frac{1}{L_{s}} & -\frac{L_{s}}{2}<z<\frac{L_{s}}{2}\;\\ 0 & other\;regions \end{array}\right.$$

This model assumes that the intracavity light is evenly confined by the grating reflectors. The fiber lasers sensor can be generally characterized by the effective cavity length *L_eff_*. The effective length is approximately equal to the grating separation for Fabry–Perot lasers, which can be expressed as *L_eff_* = 1/*κ* for the short structure. 

The sensor responses are determined by the integration of laser mode distribution ${\left|e\left(z\right)\right|}^{2}$ over the interaction length *L**eq*. Because *L**eq* is much shorter than the laser cavity, the sensitivity curve is approximated as the laser intensity profile along the fiber cavity. We changed the laser mode distribution via different cavity lengths, *L**s*. Figure 9a plots the PA intensities along the fiber at different cavity lengths, where the source-to-fiber distance was 250 μm. The sensors showed flat-top profiles if the cavity lengths were 3 mm and 5 mm, whereas a Gaussian-like profile appeared in the 2 mm one. Also, the 2 mm one showed higher sensitivity due to its confined laser mode. The full width at half maximum of these sensors was calculated as 2.2 mm, 3.5 mm, and 4.6 mm, respectively. 

When the acoustic pressure is along the principal axis, the induced birefringence variation can be maximized. At 45°, the acoustic pressure induces nearly equal phase changes for both polarization modes, thus the beat signal produces nearly zero frequency shift. Figure 9b shows the azimuthal transformation of the fiber sensitivity. It was measured by the sensor rotating at 10° per step, maintaining a fixed acoustic point source. The angular response exhibits a |cos(2*θ*)| profile, which offers a 60° full angle at half maximum along this direction. 

## 4. Photoacoustic Microscopic Imaging

The dual-polarized fiber laser sensor was used to develop OR-PAM, as shown in Figure 10. The optical beam from a 532 nm pulsed laser (VPFL-G-20, Spectral Physics, Santa Clara, CA, USA) with 1.8 ns pulse width and ~100 nJ pulse energy was collimated, reflected and then focused on the sample surface. The excited PA signals were detected by the fiber laser sensor, and the optical signal was measured by the photodetector and digitalized by a data acquisition card (see Figure 2 for details). To maximize the detection sensitivity, the optical focus and the fiber laser sensor were carefully aligned in the water tank. Meanwhile, the optical beam was carefully positioned to avoid being blocked by the optical fiber.

To quantify the lateral resolution of the OR-PAM system, a sharp blade edge was linearly scanned with a step size of 0.18 μm. The lateral resolution was measured as 3.20 μm. To validate its imaging field of view (FOV), the Galvo mirror controlled laser beam was raster-scanned over a black tape with spatially uniform absorption. The fiber laser sensor was fixed 1.60 mm above it. The restored maximum intensity projection (MIP) image was over 3 × 3 mm^2^, and the −6 dB FOV was calibrated as ~3 × 1.6 mm^2^. The penetration depth of the photoacoustic microscopy (PAM) was estimated to be ~800 μm in the phantom experiment and ~200 μm for the in vivo imaging [43,44,45], as determined by the numerical aperture and laser wavelength. To test the sensor’s stability, two human hairs were imaged in B-scan mode over 30 min. The peak to peak amplitude of the PA signal is shown in Figure 11. No noticeable noise variation of the sensor was observed. In a previous report [43], the fiber laser sensor remained stable while the sensor was scanned at ~10 mm/s in water due to the heterodyning detection.

In vivo experiments on the mouse ear and brain were conducted to demonstrate the setup’s imaging capability. Here, the laser pulse repetition rate was 100 kHz and the scanning rate along the slow and fast axis were 0.2 and 100 Hz, respectively. It took 5 seconds to capture a 3D image. The laser pulse energy on the tissue surface was 300 nJ and the imaging FOV was ~2 × 2 mm^2^. Figure 12a,b exhibits the MIP and 3D images of the mouse ear, where both trunk vessels and capillaries can be resolved clearly in the PA microscopic image. The restored mouse brain images with and without skull are presented in Figure 12c,d, where certain amounts of capillaries become much clearer without the skull. Therefore, the fiber laser sensor exhibited excellent performance in OR-PAM, being able to detect PA signals with great stability and high sensitivity.

## 5. Conclusions

In this work, recent developments in dual-polarized fiber laser ultrasound sensors for application in optical-resolution photoacoustic microscopy were reviewed. The fiber laser sensor presented herein demonstrated excellent characteristics, such as high sensitivity, broad bandwidth, minimized size, and great stability. While the photoacoustic waves exerted pressure and induced harmonic vibration of the fiber, the frequency shift of the beating signal between the two orthogonal polarization modes could be captured efficiently. Specifically, a 60 µm fiber laser could achieve an NEP of 40 Pa over a 50 MHz bandwidth. Note that the NEP can be further improved separately via using either a high-power photodetector or averaging multiple duplicated optical signals to suppress the noise from the light source, optical amplifier, photodetector, and data acquisition card. Meanwhile, the frequency shift of the beat signal coming from the dual-polarization mode resisted external perturbations without any frequency-locking techniques. As a result, OR-PAM based on the fiber laser sensor was developed and calibrated, achieving a lateral resolution of 3.2 µm and a FOV of 3 × 1.6 mm^2^. Moreover, excellent in vivo results of photoacoustic imaging in the mouse ear and brain were presented, wherein microvasculature can be clearly visualized. Therefore, the fiber laser ultrasound sensor offers a new tool for all-optical photoacoustic imaging. Moreover, the miniature size and side-looking manner give it the potential for photoacoustic endoscopy. 

## Figures and Tables

**Figure 1 sensors-19-04632-f001:**
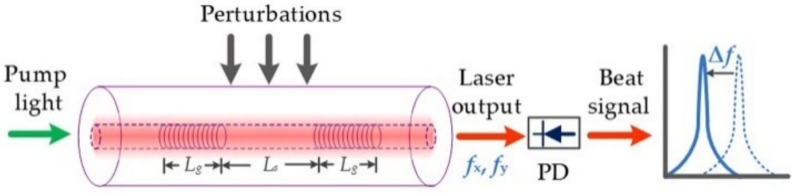
Principle of the dual-polarization fiber laser ultrasound sensor. PD, photodetector.

**Figure 2 sensors-19-04632-f002:**
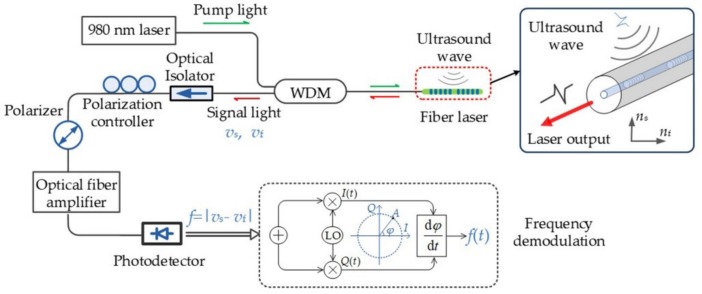
Ultrasound wave detection and signal demodulation based on the fiber laser sensor. Figure adapted with permission from Ref. [43].

**Figure 3 sensors-19-04632-f003:**
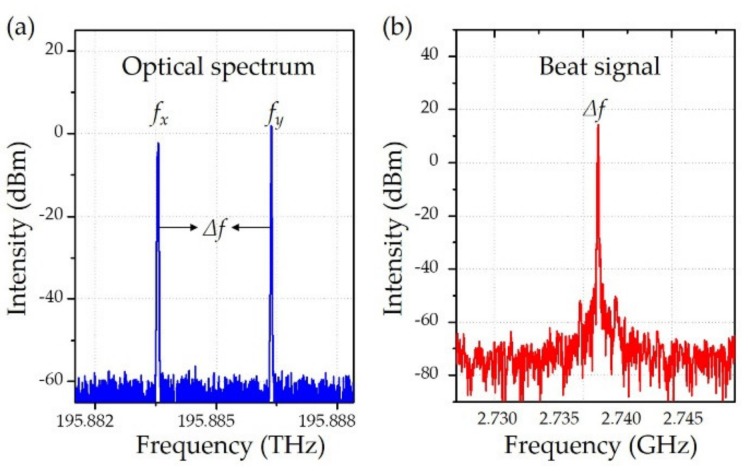
Output optical and radio-frequency spectrum of the fiber laser sensor. (**a**) Spectrum of two polarization modes measured with an optical spectrum analyzer. (**b**) Spectrum of beat signal measured by a radio-frequency (RF) analyzer after the photodetector. Figure adapted with permission from Ref. [43].

**Figure 4 sensors-19-04632-f004:**
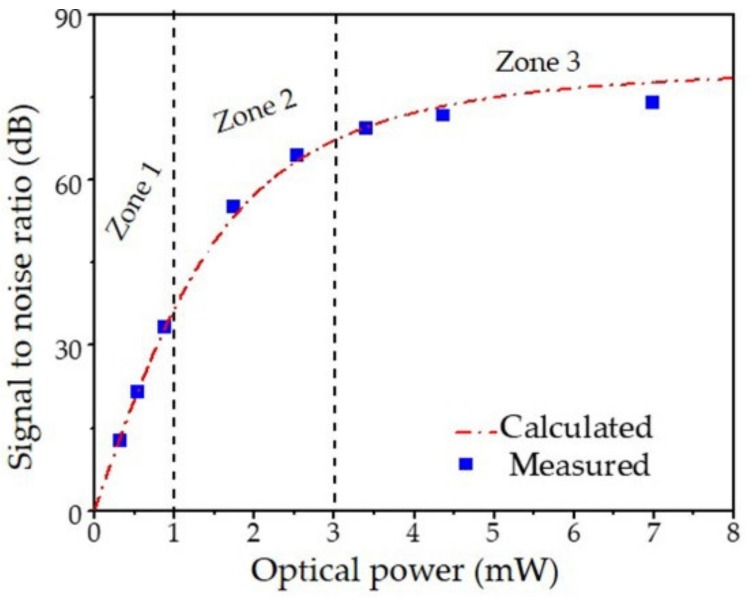
Measured and calculated signal to noise ratio (SNR) at indicated optical power at the photodetector.

**Figure 5 sensors-19-04632-f005:**
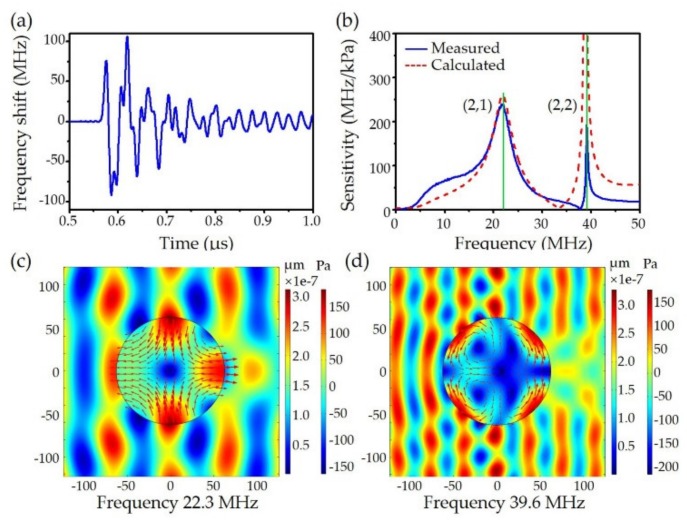
Frequency response of the fiber laser ultrasound sensor. (**a**) Transient response to a pulsed plane wave; (**b**) measured and calculated frequency responses; and (**c**) and (**d**) calculated displacement of the excited fiber vibration at the (2, 1) and (2, 2) modes.

**Figure 6 sensors-19-04632-f006:**
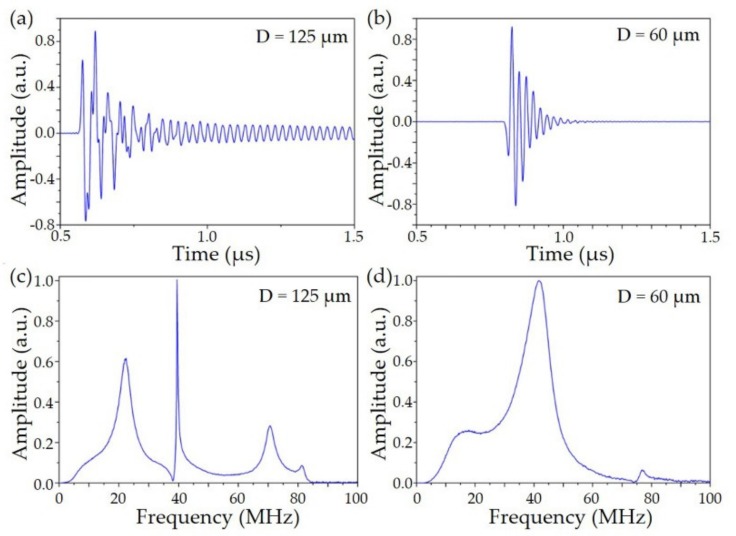
Transient response to a pulsed planar wave with different fiber diameter. Time domain responses of the 125 μm (**a**) and 60 μm (**b**) fiber sensor. Frequency responses of the 125 μm (**c**) and 60 μm (**d**) fiber sensor.

**Figure 7 sensors-19-04632-f007:**
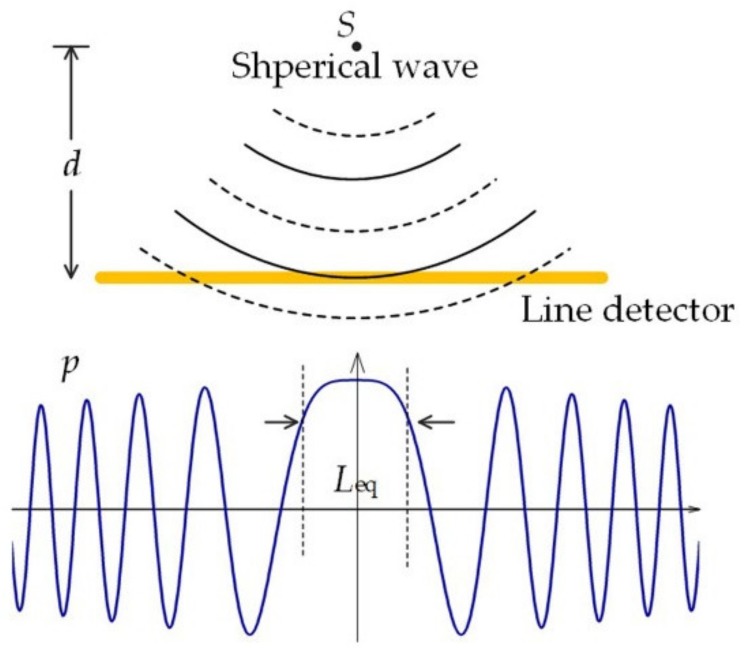
Schematic of the fiber laser sensor as an ideal line detector. A point source emits a spherical wave and the detected acoustic pressure distribution along the line detector. Figure adapted with permission from Ref. [45].

**Figure 8 sensors-19-04632-f008:**
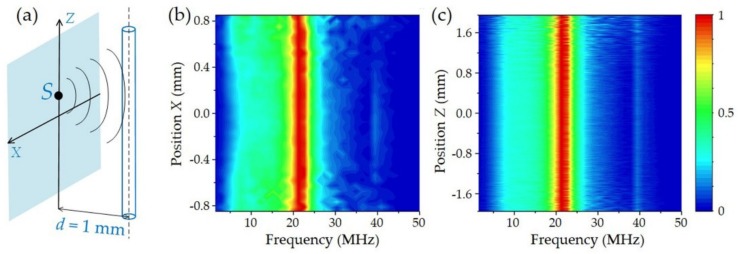
Measured frequency response of the fiber sensor. (**a**) Schematic of scanning acoustic source to measure the frequency response. (**b**) Measured frequency response with scanning source along the *x* axis (**c**) Measured frequency response with scanning source along the *z* axis. Figure adapted with permission from Ref. [45].

**Figure 9 sensors-19-04632-f009:**
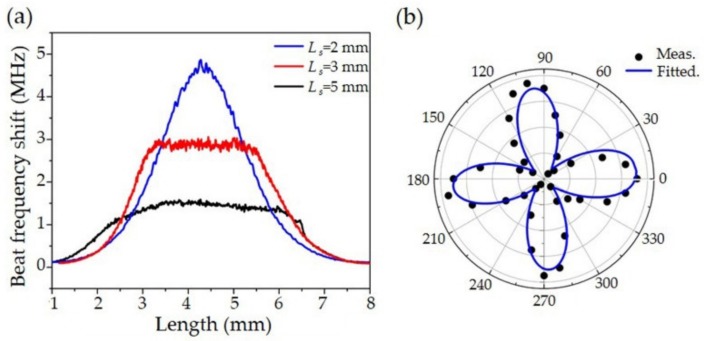
Measured acoustic responses along (**a**) longitudinal and (**b**) azimuthal direction. Figure adapted with permission from Ref. [45].

**Figure 10 sensors-19-04632-f010:**
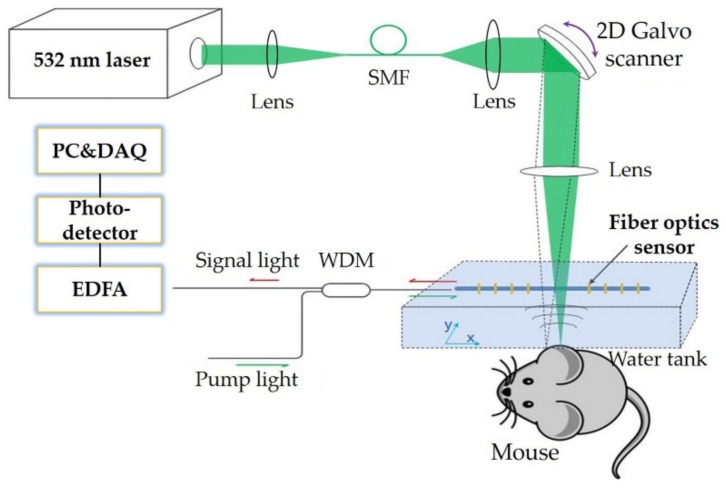
The optical resolution photoacoustic microscopy (OR-PAM) experimental setup, based on the fiber laser sensor. SMF: single mode fiber; WDM: wavelength-division multiplexer; EDFA: erbium-doped fiber amplifier; DAQ: data acquisition. Figure adapted with permission from Ref. [44].

**Figure 11 sensors-19-04632-f011:**
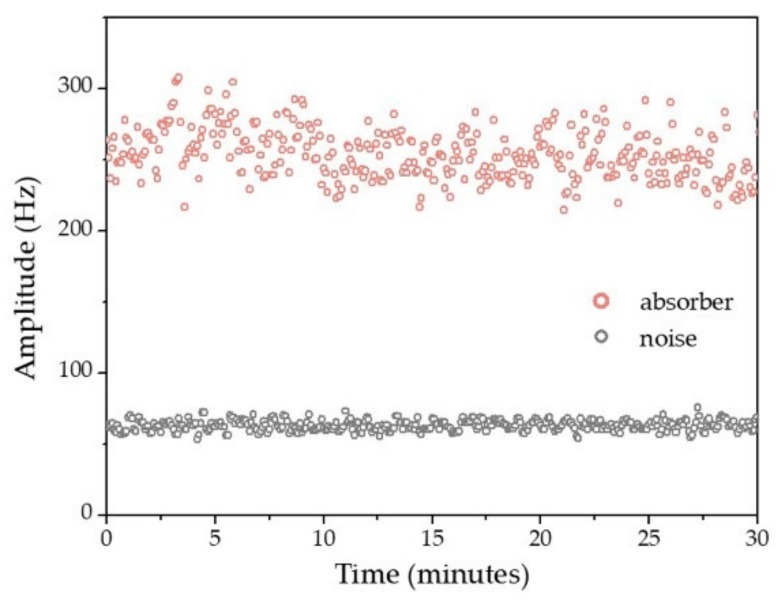
Photoacoustic (PA) amplitude extracted from the B-Scan maximum intensity projection (MIP) image of human hairs as absorbers for 30 min.

**Figure 12 sensors-19-04632-f012:**
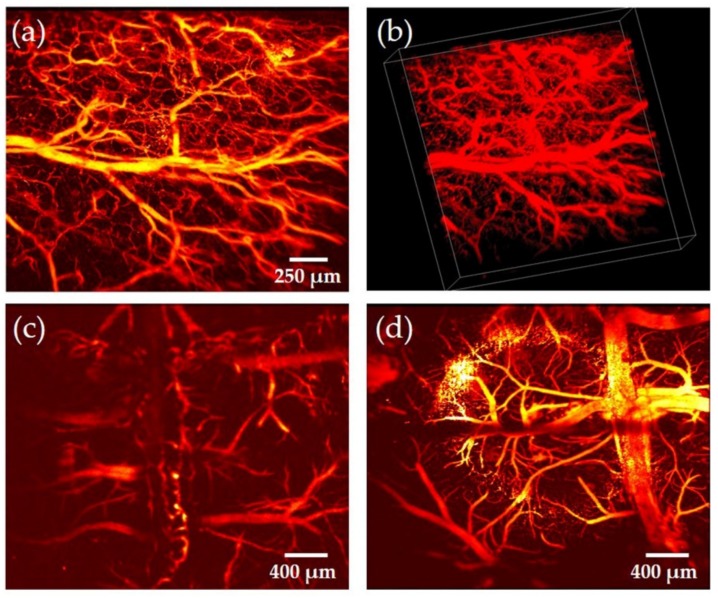
In vivo OR-PAM of the mouse ear and brain using the fiber laser sensor. (**a**) 2D MIP imaging of the mouse ear. (**b**) 3D volumetric image of the mouse ear, the white box is 2.2 × 2.2 × 0.52 mm. (**c**) In vivo imaging of the mouse brain with the skull. (**d**) In vivo imaging of the mouse brain without the skull. (a) and (b) adapted with permission from Ref. [44].

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
