# Peer review of "Dual-Polarized Fiber Laser Sensor for Photoacoustic Microscopy"

_sensors, 2019, doi:10.3390/s19214632_

Round 1
Reviewer 1 Report
Dear Authors,
you presented that optical-resolution photoacoustic microscopy (OR-PAM) provides high-resolution, label-free and non-invasive functional imaging for a broad range of biomedical applications. You presented characterization and sensitivity optimization of this type of sensor. I think that the manuscript is not suitable for publication in this form as a review, though. You must decide if you want to publish an article or a review. Below is a list with comments:
If you want your manuscript to be a review, you need more references to present a more thorough and deeper description of the subject. If this is a review, then in the ‘Principle of dual-polarized fiber laser sensor’ please present methods that other groups applied and compare them. The same should be done in other sections. In my opinion, the paper in its present form is not prepared as a Review. I suggest you resign from a Review paper and prepare a manuscript as a regular article describing your own research, as you do in the text. If this is an article you should: add more information in the introduction section. At this moment I can see only one sentence about your motivation. correct the quality of figures 2,3,4 and 5. add the city and county of production of the devices. add references to the formulas and explain all used variables.
3) The manuscript needs a thorough English correction in respect of grammar and sentence construction, punctuation and typos.
In conclusion, the manuscript should undergo a substantial revision and the definition of what you write before being re-submitted.
Reviewer 2 Report
The article makes an impression of being innovative and represent a relevant improvement for the state of the art. However, I suggest to add more mathematical details regarding the cavity behaviour.
Reviewer 3 Report
Please see attached file.

Round 2
Reviewer 1 Report
The article includes sufficient technical data, a description of results, and conclusions. It includes appropriate references and it is correct linguistically.
The paper is ready for publication.